# Antioxidant and Cytoprotective Effect of *Piper aduncum* L. against Sodium Fluoride (NaF)-Induced Toxicity in Albino Mice

**DOI:** 10.3390/toxics7020028

**Published:** 2019-05-16

**Authors:** Oscar Herrera-Calderon, Luz Chacaltana-Ramos, Ricardo Ángel Yuli-Posadas, Bertha Pari-Olarte, Edwin Enciso-Roca, Johnny Aldo Tinco-Jayo, Juan Pedro Rojas-Armas, Luis Miguel Visitación Felix-Veliz, Cesar Franco-Quino

**Affiliations:** 1Faculty of Pharmacy and Biochemistry, Universidad Nacional Mayor de San Marcos, Lima 15001, Peru; lfelixv@unmsm.edu.pe; 2Faculty of Pharmacy and Biochemistry, Universidad Nacional San Luis Gonzaga, Ica 11001, Peru; luzjos934@gmail.com (L.C.-R.); berthapari@hotmail.com (B.P.-O.); 3Universidad Continental, Huancayo 12002, Peru; ryuli@continental.edu.pe; 4Faculty of Health Sciences, Universidad Nacional de San Cristóbal de Huamanga, Ayacucho 05001, Peru; encisoqf@hotmail.com (E.E.-R.); aldotinco2@hotmail.com (J.A.T.-J.); 5Faculty of Medicine, Universidad Nacional Mayor de San Marcos, Lima 15001, Peru; jprojasarmas@yahoo.com; 6Faculty of Dentistry, Universidad Nacional Mayor de San Marcos, Lima 15001, Peru; cesar.franco1@unmsm.edu.pe

**Keywords:** *Piper aduncum*, antioxidant, cytoprotective, micronucleus test, Comet assay

## Abstract

*Piper aduncum*, commonly known as matico, is a plant that grows in the mountainous and coastal regions of Peru, and is studied for its antimicrobial properties and various ethnopharmacological uses. The main objective of this study was to determine the cytoprotective and antioxidant effects of the methanolic extract of *Piper aduncum* leaves in *Mus musculus* previously administered with sodium fluoride (NaF) using the Micronucleus test and the Comet assay. The extract was administrated orally in four different concentrations: 150, 300, 600, and 1200 mg/Kg for ten days. At the 11th day, a single dose of NaF was administrated via intraperitoneal at 20 mg/Kg. The genotoxicity study was performed with mice from the strain BALB/c, using the Micronucleus test on bone marrow and the Comet assay on peripheral blood according to OECD guidelines 474 and 489, respectively. The statistical analysis was performed by median analysis with ANOVA. Significant differences were found in Micronucleus frequency between the highest concentrations of *Piper aduncum* and NaF. The Comet assay showed significant reduction of NaF-induced damage on erythrocytes depending on the different concentrations of the extract which were evaluated in this study. It is concluded that the methanolic extract of *P. aduncum* leaves has cytoprotective and antioxidant activity against sodium fluoride.

## 1. Introduction

*Piper aduncum* L. *(P. aduncum*) is known as “matico”, in different places from Peru and widely used in the traditional medicine of Latin-America. The Piperaceae family, in which the genus Piper belongs, comprises approximately 2000 species distributed in the tropical regions worldwide [1]. *P*. *aduncum* is used in folk medicine as an anti-inflammatory, for wound healing, treating rheumatic afflictions and diarrhea, and as an antiseptic. [2] Previous phytochemical studies of *P. aduncum* reported the isolation of chalcones, dihydrochalcones, flavanones, chromene, phenylpropanoids, and benzoic acid derivatives. Some of them are shown in Figure 1, which have been studied as cytotoxic, antimicrobial, and insect repellent agents [3,4]. 

Pharmacological effects of *P. aduncum* extracts (ethanol and methanol) and its essential oil have been demonstrated, including antileishmanial, antibacterial, cytotoxic, and antifungal activities [5]. Furthermore, isolated compounds from leaves have been shown to be very active against promastigote and intracellular amastigotes, causing damaging effects on DNA, and have antileishmanial, antimicrobial, molluscicidal, antitumor, and antifungal activities [6,7]. Otherwise, it has been demonstrated that some species of Piper have antitumor activity, such as *Piper claussenianum*, which has an in vitro effect on breast cancer tumor cell lines (MCF-7). *P. longum* and *P. nigrum* have been evaluated as immunomodulatory, antioxidant, and antiproliferative agents in breast cancer in vitro [8]. Other studies tested the essential oil of *P. aduncum* leaves and have evidenced its variety of uses as a molluscicide, insecticide, and antibacterial agent [9]. 

Sodium fluoride (NaF) is an inorganic salt used commonly on dental treatments to prevent caries, topically, or in some cases in municipal water fluoridation systems. It has also been used as an insecticide, wood preservative, and in the manufacture of glass. It has also been used in cytotoxic research models. In high doses it has a very potent oxidative effect which causes direct damage to DNA molecules. For example, it can break DNA strands and cause apoptosis in human cells, such as erythrocytes, and generate oxidative stress on oral mucosa from rats. Among the main adverse reactions is the generation of free oxygen radicals in cells [10]. Studying genotoxicity can done via biological models such as the Micronucleus (MN) test and the Comet assay. The mammalian in vivo MN test’s purpose is to identify substances that cause cytogenetic damage, which results in the formation of micronuclei on the studied cells, in this case the erythrocytes [11]. The chosen cells are studied in the bone marrow, because their maturing progress consists of expulsing the nucleus; therefore, any damage to the DNA can be seen in this process, especially because the mature kind lacks a nucleus. The in vivo alkaline Comet assay is used to detect DNA strand breaks in cells under alkaline conditions (pH > 13); these breaks can be the result of direct interactions with DNA, sites of DNA that are alkali labile, or as a consequence of transient DNA strand breaks resulting from DNA excision repair [12]. 

Peruvian flora is considered one of the best sources of curative plants in the world [13] Furthermore, various species with potential use for commercial purposes have not been studied, and until today there previous study on the cytoprotective effect of *P. aduncum* from Peru. Environmental conditions can alter phytochemical production, thus plants from different regions may have different profiles. 

Although, a wide number of phytocompounds (including alkaloids, phenolic compounds, glycosides, flavonoids, anthocyanins, etc.) may have cytotoxic and genotoxic properties, the phytochemicals of Piper species, in addition to having a long history of use, their natural origin and widespread use do not guarantee their safety. Therefore, the aim of this study was to determine the cytoprotective and antioxidant effects of the methanolic extract of *P. aduncum* leaves in albino mice previously administered with sodium fluoride (NaF) using the Micronucleus test and the Comet assay.

## 2. Materials and Methods

### 2.1. Chemicals

The materials applied in this research are as follows: 2-thiobarbituric acid (TBA), gallic acid, quercetin, FeCl_3_, MgCl_2_, Triton X-100, NaCl, Tris, EDTA, Boric acid, and DMSO from Merck; Folin–Ciocalteu reagent (FCR), 2,4,6-tripyridyl-striazine (TPTZ), 2,2′-azino-bis-3-ethylbenzthiazoline-6-sulphonic acid (ABTS) and 1,1-diphenyl-2-picrylhydrazyl (DPPH) radical and all chemicals used in our study were of analytical or reagent grade (Sigma-Aldrich, St. Louis, MO, USA).

### 2.2. Plant Material

The leaves of *P. aduncum* were collected from Huamanga, Ayacucho, Peru in January 2018 and taxonomically identified by the staff at the Herbarium from Universidad Nacional Mayor de San Marcos (Lima, Peru).

### 2.3. Experimentation Animals

For the acute oral toxicity experiment: six male Hotzman albino rats weighing 180–190 g (6–8 weeks old) were divided in two groups, one being the control. For feeding, balanced rodent special food was used and animals had access to water ad libitum. Physical conditions were: temperature at 19.6 °C, humidity 67%, and 12 h light/dark natural cycle.

For the micronucleus test and comet assay: 30 healthy, albino mice *Mus musculus* (BALB/c) between 6–10 weeks olds, weighing 25–35 g were supplied by the Instituto Nacional de Salud (Lima) seven days before the experiments. Animals were brought to the animal facilities of the university and kept in boxes in air-conditioned room (23 °C) and relative humidity of 50%, with a 12-h dark/light cycle and were fed with special balanced pellet (Instituto Nacional de Salud, Peru) and water ad libitum.

All protocols were approved by the Ethics Committee of the Faculty of Medicine, UNMSM, Act 0310 (approval date: 4 November 2017) in accordance with the Guide for the Care and Use of Laboratory Animals.

### 2.4. Extract Preparation

The dried leaves of *P. aduncum* were ground to powder form. The powder was soaked for 72 h in an appropriate volume of methanol to obtain 100 mg/mL solution. The solution was evaporated by using a rotavap in order to obtain a dried extract, then it was stored in a brown vial at 4 °C until further use [14].

### 2.5. Antioxidant Activity

To determine the antioxidant capacity, three tests were performed: 

#### 2.5.1. Ferric Reducing ability of plasma (FRAP)

FRAP assays were performed using the technique by Benzie and Strain (1996) modified by Rao et al. [15]. One mL of the FRAP substance was mixed with 1 mL distillated water and the extract of *Piper aduncum*, then exposed at 37 °C for 15 minutes and read on the spectrophotometer (539 nm) compared with a standard curve with different concentrations of Fe^+2^.

#### 2.5.2. 2-Diphenil-1-picrilhidracil (DPPH)

The technique used was proposed by Brand-Williams et al. (1995) [16]. The assay mixture consisted of various concentrations of extract at 1 mL of acetate buffer pH 6 (0.1M), methanol, and 0.5 mL of DPPH 0.1 mM, mixed and heated at 37 °C for 30 min to be read on spectrophotometer at 517 nm. The control was made comparing with the solution of DPPH without the extract. 

#### 2.5.3. Thiobarbituric Acid Reactive Substances (TBARS)

TBARS assays were performed according the technique described by Draper (1990) [17] which evaluates the lipoperoxidation by measuring the formation of malondialdehyde (MDA) using the reactive acid thiobarbituric in the presence of acid trichloroacetic, then it was read with a spectrophotometer at 532 nm. 

### 2.6. Determination of Flavonoids and Polyphenols

According to the technique by Wolfe et al. (2003) [18] for flavonoids determination, 205 µL of methanolic extract was mixed with 125 µL water and 75 µL sodium nitrate at 5%. It was set for 5 min and then aluminum trichloride was added at 10%. The mixture was set again for 6 min and 0.5 mL 2 M sodium hydroxide and 275 µL water were added, then the mixture was measured by spectrophotometer at 510 nm. The technique for polyphenols determination was from Herrera et al [1]. For the mixture, 1 mL of Folin–Ciocalteu solution at 10% and 0.1 mL of the extract were set for 5 min, then added 1 mL of sodium carbonate at 7.5%. After 15 minutes, it was read by spectrophotometer at 725 nm. 

### 2.7. Acute Oral Toxicity

According to OECD guideline 423 for testing chemicals [19], the animals were treated and observed for 14 days. The treated group (three rats) were fed with the extract at a concentration of 2000 mg/Kg/day using an intragastric cannula and the control group (three rats) with 1 mL distillated water. The animals were weighed at the beginning of the experiment, at the 7th day, and at the end. Then, they were sacrificed by cervical dislocation for macroscopic organ analysis. During the entire experiment (14 days), abnormal behavior was recorded.

### 2.8. Genotoxicity Study

Following OECD guidelines 474 for MN and 489 for Comet assay [20], the treatment lasted ten days, and the 30 animals were divided into six groups of five mice each (three males and two females): the negative control group was administrated orally with the diluent, DMSO 10%, and no bioactive substances; the extract was administrated through an intragastric cannula at different doses of 150 mg/Kg, 300 mg/Kg, 600 mg/Kg, and 1200 mg/Kg; the positive control group did not receive any substance until day 11, when all groups except the negative control were administrated with a single dose of 20 mg/Kg NaF via intraperitoneal injection. The animals were observed for one more day until day 12, when they were sacrificed by cervical dislocation.

#### 2.8.1. Micronucleus Test

After the sacrifice, the femur was removed for the extraction of bone marrow with physiological serum enriched with BSA 10% and centrifuged at 1500 rpm, the supernatant was disposed, and the cellular sample was smeared and left to be dried and fixed with cold methanol. The slides were stained with Giemsa 2% for the microscopic analysis at 1000× using a Nikon Eclipse 50i microscope. OECD guideline 474 establishes the identification of MN cells: a principal nucleus and one or more nuclear structures named MN, the MN is rounded or almond shaped and is 1/3–1/16 from the main nucleus, it has the same intensity, texture, and focal plane as the nucleus. Two-thousand polychromatic erythrocytes (1000 cells per replicate) were scored for each sample. 

In addition, 500 cells (polychromatic and normochromic erythrocytes) were scored to determinate the cytotoxic index (CTI). This index is used to evaluate the number of cells with DNA damage in comparison with the ones that did not present DNA alterations and could mature, under the premise that genotoxic substances alter the polychromatic erythrocytes.
$$\mathrm{Cytotoxic}\;\mathrm{index}\;(\mathrm{CTI})\;=\frac{polychromatic\;erythrocyte}{normochromic\;erythrocyte}$$

#### 2.8.2. Comet Assay

The comet assay was performed using the Guidelines of OECD Test 489 Alkaline Comet assay with minor modifications. Approximately 40 µL of blood was resuspended in 110 µL of 0.5% low melting point agarose (LMA), smeared on microscope slides precoated with 100 µL of 0.5% normal melting point agarose (NMA) previously dried at 65 °C and covered with a coverslip and kept at 4 °C until solidification. The coverslips were removed and cells were lysed for 2 h at 4 °C in a dark chamber containing a cold fresh lysing solution. To allow DNA denaturation, unwinding, and exposure of alkali-labile sites, the slides were placed in a horizontal gel electrophoresis tank filled with cold electrophoresis solution for 20 min. Electrophoresis was performed in the same solution for 20 min at 0.73 V/cm and 300 mA. Then, the slides were neutralized with washes for 2–5 min with 0.4 M Tris (pH 7.5), fixed with cold absolute ethanol for three min and stored in the dark at room temperature. Before the microscope analysis, the slides were stained with 50 µL of Hoescht 33258 (50 ug/mL), then observed at 400× magnification at a Nikon Eclipse 50i fluorescence microscope. One hundred randomly selected cells (50 cells from each of the two replicate slides) were analyzed per sample. These cells were visually analyzed according to classes, ranging from undamaged, score 0 (completely undamaged; 0 × 100 cells) to highly damage, score 400 (completely damaged; 4 × 100 cells). On the other hand, to visualize DNA damage, slides were examined at 400× and 100 cells were randomly selected for analysis in each sample. The tail DNA (%) and tail moment were used to measure DNA damage because they give the most meaningful results in genotoxicity studies [21]. 

### 2.9. Statistical Analysis

For the antioxidant activity, the tests results were presented as medians from three determinations ± standard deviation (SD). The differences between medians were determined by Kruskall–Wallis test. The correlations between variables were made by the Pearson test and performed using SPSS. 

For the genotoxicity study, statistical analysis of medians of the different groups of the experiment was made using ANOVA within SPSS. The statistical significance level was 95%, *p* < 0.05.

The percentage reduction reflected the reduction of damage in comparison with the positive and negative control, showing more objectively the cytoprotective effect of the plant extract.
$$\mathrm{Reduction}\;(\%)\;=\left[\frac{A-B}{A-C}\right]\times 100$$

A: the median of damage in positive control; B: the median of damage in the treatment group *(Piper aduncum* + CP); C: the median of damage in the negative control.

## 3. Results

### 3.1. Antioxidant Activity

The results found for the methanolic extract of *Piper aduncum* leaves are shown in Table 1. 

Total polyphenols estimated using the Folin–Ciocalteu method were found in a lower amount (16.09 mg GAE/g). Flavonoid contents were also found to be minor (3.58 mg CE/g). Although, Herrera et al. (2018) [1] reported 311.11 mg GAE/g and 154.98 mg RUE (rutine equivalents) for total polyphenols and flavonoid content, respectively, in leaf ethanolic extract from *Piper aduncum* L. They concluded that their results indicate the position and/or the number of glycosyl groups present in the molecule play a significant part in the antioxidant activity. 

### 3.2. Acute Oral Toxicity Study

Oral administration of *P. aduncum* extract did not produce any mortality or abnormal behavioral response in rats during the 14 days of the experiment. In the anatomopathological study, there were no organ or system alterations compared to the control group. The lethal dose 50 (LD_50_) for this extract appeared to be >2000 mg/kg. 

### 3.3. Genotoxicity Study

According to the results shown in Table 2, treatments of 150, 300, 600, and 1200 mg/kg body weight *P. aduncum* extract caused a significant decrease in the MNPCE frequency by 52.4% (9.05 ± 0.707), 67.9% (6.42 ± 1.817), 83.3 % (3.81 ± 0.837), and 97.6% (1.42 ± 0.894), respectively, compared to NaF exposure (17.81 ± 6.140), showing the protective capacity of the plant extract. Also, the results of the present study showed that the leaf extract of *P. aduncum* can mitigate the cytotoxicity of NaF and promotes erythropoiesis in mice.

Like the results of the Micronucleus test, the data from Comet analysis of extraction of bone marrow erythrocytes in mice showed that all the doses evaluated (*P. aduncum* + NaF) were capable of reducing the NaF-induced DNA damage.

## 4. Discussion

This study aimed to determine the antioxidant capacity of the methanolic extract of *P. aduncum* and was performed using the most important markers, such as TBARS which measures the lipoperoxidation, meaning damage to lipid substances by free radicals; DPPH which measures specific free radicals; and FRAP which measures the antioxidant effect by reduced iron ions. The extract demonstrated that it has very high levels of these markers; therefore, it has a high antioxidant activity related to the high concentration of polyphenols and flavonoids. These markers were compared with studies from Indonesia [24], and they turned to have greater antioxidant capacity than the extracts from other studies. 

The evaluation of acute oral toxicity showed a LD_50_ of >2000 mg/kg, which comes as no surprise because it is a solution almost purely composed of flavonoids and polyphenols, and it has been widely used by the native people by thousands of years with no apparent adverse reactions. This LD_50_ ranks in the 5th category of the OECD which corresponds to extremely low toxicity and has low risk of toxic reactions when orally administrated. 

The Micronucleus test and the Comet assay are two of the most reliable markers for mutagenic and teratogenic activity in cells with high mitotic activity. They show chromosomic damage, and therefore genotoxicity, in this case in immature erythrocytes exposed to substances like sodium fluoride. These markers also provided information about the protector effect which some substances have on the DNA. The results show that as the concentration of the extract rises, the Micronucleus frequency and the damaged nucleotides drops. Therefore, the methanolic extract of *Piper aduncum* can protect DNA from damage caused by some substances. As shown by the cytotoxic index, which shows the capability of the cell to mature, more cells mature when they have higher concentration of the extract, meaning there was no DNA damage, either because of successful reparation or because there was no damage at all. These benefits can be explained because of the extract’s high antioxidant activity which can offset the oxidant effect of sodium fluoride, which otherwise alters DNA by producing free radicals. Furthermore, it is known that fluoride increases caspase-3, caspase-8, caspase-9, and bax protein expression, and reduces bcl-2 protein expression in rodent liver [25]. 

In future studies, it is recommended to elucidate the mechanism of the methanol extract of *P. aduncum* in order to understand its antioxidant and cytoprotective effects.

## 5. Conclusions

In conclusion, the methanolic extract of *Piper aduncum* has a cytoprotective effect against mutagenic substances such as sodium fluoride. Also, this extract has proven to have very high antioxidant activity, a high concentration of flavonoids and polyphenols, and a very low index of acute oral toxicity, with a LD_50_ of >2000 mg/kg. 

Studying of antimutagenic substances should be performed widely, especially with products of low toxicity and low or non-adverse reactions. Therefore, a more specific study should be performed to isolate the exact components of this extract.

## Figures and Tables

**Figure 1 toxics-07-00028-f001:**
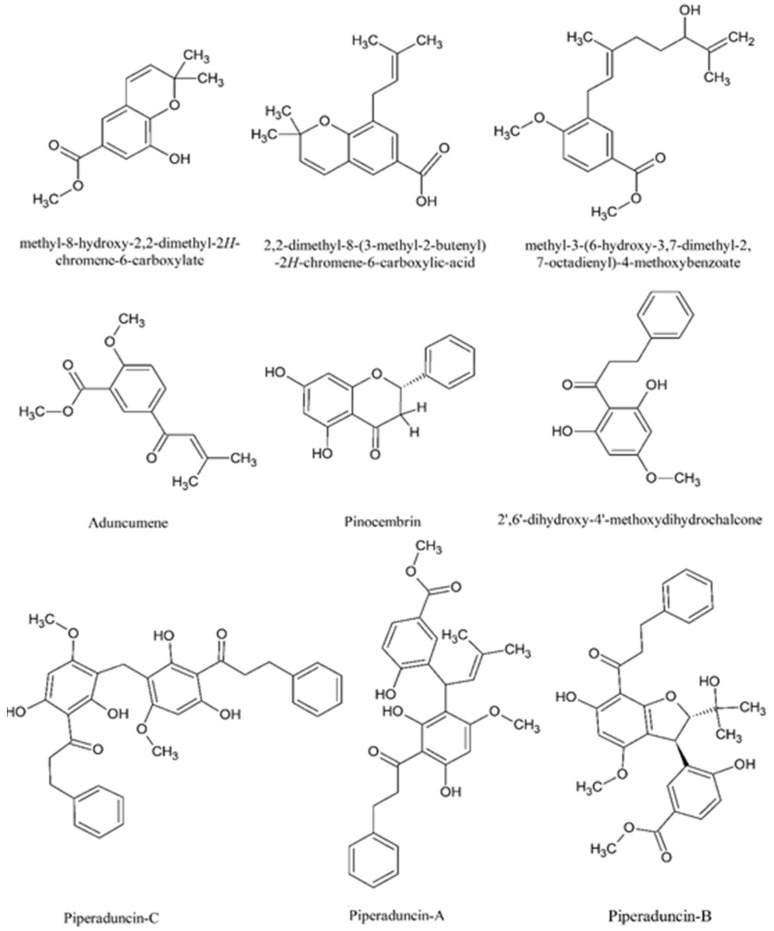
Main phytochemicals found in *Piper aduncum* L. leaves.

**Table 1 toxics-07-00028-t001:** Bioactive compounds and antioxidant activity of methanolic extract of *Piper aduncum* leaves.

Methanolic Extract	Polyphenols (mg GAE/g)	Flavonoids (mg CE/g)	FRAP (µmol Fe^+2^/g)	DPPH (µmol TEAC/g)	TBARS (µg MDA/g)
	19.00 ± 0.8	4.10 ± 0.11	40.63 ± 0.65	0.790 ± 0.5	260.0 ± 0.56
**Other reports of antioxidant activity**	4.94 ± 0.05 (ethanol extract) [22]	3.8 ± 0.03 (n-hexane extract) 6.7 ± 0.02 (ethyl acetate extract) 8.3 ± 0.01 (ethanol extract) [22]	121.03 ± 0.27 (µg ascorbic acid equivalent/mL) [22]	1248.82 ± 17.14 µg/g (n-hexane extract) 1454.7 ± 0.38 µg/g (ethyl acetate extract) 129.54 ± 0.41 µg/g (ethanol extract) [22]	
				85.24 ± 0.60 (ethanol extract) [23]	

GAE—gallic acid equivalents; CE—catechin equivalents; Data represented as means ± SD (*n* = 3).

**Table 2 toxics-07-00028-t002:** Genotoxicity study: Micronucleus test and Comet assay.

Experimental Group	Micronucleus Test			Comet Assay			
	MNPCE/1000PCE (median ± SD)	CTI PCE/NCE	Reduction (%)	Number of Cells Expressed as AU	Reduction (%)	Tail DNA (%)	Tail Moment (%)
***PA* 150 mg/Kg + PC**	9.05 ± 0.707 ^a,b^	2.041 ± 0.408	52.4%	104.23 ± 10.93 ^a,b^	65.58%	0.15 ± 0.01	0.14 ± 0.01
***PA* 300 mg/Kg + PC**	6.42 ± 1.817 ^a,b^	3.27 ± 1.13	67.9%	82.40 ± 22.54 ^a,b^	75.4%	0.17 ± 0.02	0.11 ± 0.05
***PA* 600 mg/Kg + PC**	3.81 ± 0.837 ^b^	7.96 ± 2.58	83.3%	69.20 ± 10.56 ^b^	82.65%	0.25 ± 0.01	0.16 ± 0.02
***PA* 1200 mg/Kg + PC**	1.42 ± 0.894 ^b^	14.69 ± 8.50	97.6%	42.40 ± 24.13 ^b^	95.9%	0.36 ± 0.005	0.20 ± 0.01
**DMSO 10% (NC)**	1.04 ± 0.707	10.66 ± 7.81	-	32.80 ± 12.56	-	0.11 ± 0.01	0.16 ± 0.01
**NaF 20 mg/Kg (PC)**	17.81 ± 6.140 ^a^	0.104 ± 0.33	-	242.60 ± 37.47 ^a^	-	8.10 ± 1.00	7.00 ± 0.50

Each treatment contains three females and two males; NC: Negative Control; PC: Positive Control; MN: Micronucleus, PCE: Polychromatic erythrocyte, NCE: Normochromic erythrocyte, CTI: cytotoxic index; AU: Arbitrary units; SD: Standard deviation; Tukey’s Parametric test. ^a^ significant difference from the negative control (*p* < 0.05); ^b^ significant difference from the positive control (*p* < 0.05).

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
