# Peer review of "Antioxidant and Cytoprotective Effect of Piper aduncum L. against Sodium Fluoride (NaF)-Induced Toxicity in Albino Mice"

_toxics, 2019, doi:10.3390/toxics7020028_

Round 1
Reviewer 1 Report
This article reports on a study of the antioxidant properties of a methanolic extract of Piper aduncum, its acute oral toxicity in rats and its effect on the genotoxic effects of sodium fluoride (NaF) in mice. The study shows that the extract is of low acute oral toxicity, and may protect against the ability of NaF to induce micronuclei and DNA strand breaks, which appears to be a novel finding. It is speculated that this may be due to antioxidant activity of Piper aduncum.
There are a number of methodological questions that could potentially be addressed. Although the OECD Test Guidelines (TGs) are referred to, these have not been strictly followed, for example both the micronucleus and comet assay TGs recommend using 5 animals of one sex or 5 of each sex, but instead 3 males and 2 females have been used. In addition, the number of cells scored is lower than that recommended in the TGs. The TG for the comet assay recommends determining the frequency of hedgehog cells, but this has not been reported. The TG also recommends endpoints such as % tail DNA (preferred), tail length and tail moment, however the scores were apparently based on classes ranging from undamage to highly damaged. More details on the scoring system used for the comet assay are needed as it is not currently clear how this was done and if it was appropriate.
It would be useful for the discussion to include suggestions for further work that could potentially be done to elucidate the mechanisms underlying the apparent cytoprotective effects of the extract and possibly confirm that they are due to an antioxidant effect - such as measuring markers of oxidative damage or including histopathology analysis.
The discussion notes that the extract had a greater antioxidant capacity compared with extracts in studies from Colombia and India - details should be provided of what these extracts were, and numerical comparison of the results reported in Table 1 with measurements with other extracts/antioxidants should be provided.
The introduction and discussion are lacking a consideration of why the results might be important - is it because P. aduncum could be administered for its chemoprotective effects? As an extract, whole plant or using a particular constituent?
There are a number of typos and other edits that are needed for clarity and to make the English read well. Some specific comments are below:
The numbers for the references do not appear to match up correctly. For example on line 36 reference 6 is cited in relation to copper, but reference 6 is the OECD TG for the micronucleus test.
Line 13-14: what is meant by '...medias analysis wtih ANOVA'. Does this mean a comparison of medians?
Line 14-15: it is stated that there were significant differences in micronucleus frequency 'between the highest concentrations of P. aduncum and NaF'. It needs to be clearer if this means this is between the combination of P. aduncum and NaF compared with NaF alone. Also, Table 2 suggests there were significant differences at all concentraitons of P. aduncum. not just the highest concentrations.
Line 15-16: The following sentence needs to be reworded for clarity: 'The Comet assay showed significant reduction in NaF induced damage according to different concentrations of the extract.'
Line 26: Reference 2 appears to relate to P. nigrum, are additional references needed for P. longum and P. claussenianum?
Line 27: '...extent variety...' needs rewording
Line 28: 'The roots and leaves tea were...' needs rewording.
Line 36: 'cupper' should be corrected to 'copper'
Line 45: '...the chosen cells are studied in the bone marrow...'. The TG states that the erythrocytes may be sampled from either bone marrow or peripheral blood.
Line 84-85: '...sobrenadant was purified by SPE...' Presume this should be supernatant? Also SPE should be spelt out.
Line 131: '...identification of MN cells: a principal nucleus and one or more nuclear structures named MN'. However in erythrocytes the main nucleus is extruded?
Line 157: '...test results were presented as medias from...' Should this be medians? Should be corrected in various places.
Line 167: treatment group is referred to as 'Sacha culantro + CP' This cannot be correct. Sacha culantro is also referred to in several other places.
Line 171: Table 1 refers to an ethanolic extract but the methods state that a methanolic extract was prepared.
Line 185: It is stated that the LD50 appeared to be > 5000 mg/kg, but the highest dose tested was apparently 2000 mg/kg bw.
Line 187: the doses of extract listed here don't match up with what was tested and listed in the methods and Table 2.
Line 192: is there any specific evidence to support the claim that the extract promotes erythropoiesis?
Table 2: Was any statistical analysis of the cytotoxicity index performed?
Table 2: What is AU (listed in the penultimate column)?
Table 2 footnotes: a and b are both stated to relate to a significant difference from the negative control. This cannot be correct.
Line 215: Should be '...LD50 > 2000 mg/kg ' (also in line 240)
Line 218: the acute oral toxicity categories are those of the GHS not that of the OECD (the OECD TG for acute oral toxicity refers to the GHS categories' and con
Author Response
Thank you for your observations Dear Reviewer, we are pleased to read your suggestions:
There are a number of methodological questions that could potentially be addressed. Although the OECD Test Guidelines (TGs) are referred to, these have not been strictly followed, for example both the micronucleus and comet assay TGs recommend using 5 animals of one sex or 5 of each sex, but instead 3 males and 2 females have been used. In addition, the number of cells scored is lower than that recommended in the TGs. The TG for the comet assay recommends determining the frequency of hedgehog cells, but this has not been reported. The TG also recommends endpoints such as % tail DNA (preferred), tail length and tail moment, however the scores were apparently based on classes ranging from undamage to highly damaged. More details on the scoring system used for the comet assay are needed as it is not currently clear how this was done and if it was appropriate.
R1: We agree that the guidelines of the OECD for Cometa and micronucleus tests were not followed, so we specify in the methodological section the word "with minor modifications". However, We decided to incorporate tail DNA and tail moment as parameters of genotoxicity.
It would be useful for the discussion to include suggestions for further work that could potentially be done to elucidate the mechanisms underlying the apparent cytoprotective effects of the extract and possibly confirm that they are due to an antioxidant effect - such as measuring markers of oxidative damage or including histopathology analysis.
R2: We improved this part as you indicated in your commentary.
3. The discussion notes that the extract had a greater antioxidant capacity compared with extracts in studies from Colombia and India - details should be provided of what these extracts were, and numerical comparison of the results reported in Table 1 with measurements with other extracts/antioxidants should be provided.
R3: Data were provided in table 1.
The introduction and discussion are lacking a consideration of why the results might be important - is it because P. aduncum could be administered for its chemoprotective effects? As an extract, whole plant or using a particular constituent?
R4; Introduction and discusion were improved according to your suggestions.
There are a number of typos and other edits that are needed for clarity and to make the English read well. Some specific comments are below:
The numbers for the references do not appear to match up correctly. For example on line 36 reference 6 is cited in relation to copper, but reference 6 is the OECD TG for the micronucleus test.
R5; English grammar was improved and references were updated.
Line 13-14: what is meant by '...medias analysis wtih ANOVA'. Does this mean a comparison of medians?
R6;The data were analyzed by using ANOVA test.
Line 14-15: it is stated that there were significant differences in micronucleus frequency 'between the highest concentrations of P. aduncum and NaF'. It needs to be clearer if this means this is between the combination of P. aduncum and NaF compared with NaF alone. Also, Table 2 suggests there were significant differences at all concentraitons of P. aduncum. not just the highest concentrations.
R6;This paragraph was modified.
Line 15-16: The following sentence needs to be reworded for clarity: 'The Comet assay showed significant reduction in NaF induced damage according to different concentrations of the extract.'
R7;This paragraph was modified.
Line 26: Reference 2 appears to relate to P. nigrum, are additional references needed for P. longum and P. claussenianum?
R8;This paragraph was modified with references updated.
Line 27: '...extent variety...' needs rewording
R9;This paragraph was modified.
Line 28: 'The roots and leaves tea were...' needs rewording.
R10;This paragraph was modified.
Line 36: 'cupper' should be corrected to 'copper'
R11;This paragraph was modified.
Line 45: '...the chosen cells are studied in the bone marrow...'. The TG states that the erythrocytes may be sampled from either bone marrow or peripheral blood.
R12;This paragraph was modified and samples were obtained from bone marrow.
Line 84-85: '...sobrenadant was purified by SPE...' Presume this should be supernatant? Also SPE should be spelt out.
R13;This paragraph was modified.
Line 131: '...identification of MN cells: a principal nucleus and one or more nuclear structures named MN'. However in erythrocytes the main nucleus is extruded?
R14;This paragraph was modified.
Line 157: '...test results were presented as medias from...' Should this be medians? Should be corrected in various places.
R15;Our results were expressed in medias no medians.
Line 167: treatment group is referred to as 'Sacha culantro + CP' This cannot be correct. Sacha culantro is also referred to in several other places.
R16;This paragraph was modified.
Line 171: Table 1 refers to an ethanolic extract but the methods state that a methanolic extract was prepared.
R17: This paragraph was modified. It is methanol extract.
Line 185: It is stated that the LD50 appeared to be > 5000 mg/kg, but the highest dose tested was apparently 2000 mg/kg bw.
R18: This paragraph was modified.the highest doses was 2000 mg/Kg.
Line 187: the doses of extract listed here don't match up with what was tested and listed in the methods and Table 2.
R18: This paragraph was modified.
Line 192: is there any specific evidence to support the claim that the extract promotes erythropoiesis?
R19: In many studies reduction of abnormal blood cells could improve erythropoiesis
Table 2: Was any statistical analysis of the cytotoxicity index performed?
R20: No statistical analysis was performed for cytotoxic index.
Table 2: What is AU (listed in the penultimate column)?
R21: AU was incorporated in legend.
Table 2 footnotes: a and b are both stated to relate to a significant difference from the negative control. This cannot be correct.
R22: This paragraph was modified.
Line 215: Should be '...LD50 > 2000 mg/kg ' (also in line 240)R23;
R23; This term was modified.
Line 218: the acute oral toxicity categories are those of the GHS not that of the OECD (the OECD TG for acute oral toxicity refers to the GHS categories' and con
R24: This paragraph was modified.
Thank you so much for your suggestions and observations in order to improve this manuscript.

Reviewer 2 Report
The manuscript titled “Cytoprotective effect of Piper aduncum against sodium fluoride (NaF) induced toxicity in mice” attempted to highlight the cytoprotective effects of Piper aduncum extract.
There are few deficiencies in the manuscripts which need to be addressed to make it publishable:
1. The extraction method is not characterized in the current manuscript and no other references were made.
2. The structures of the chemical constituents of Piper aduncum crude extract will be helpful as a figure
3. It might be a good idea to elaborate more on the known effects of Piper aduncum extract
4. The paragraphs throughout the manuscript are not well developed. Authors need to make effort to develop the manuscript and use proper transition.
5. Page 6- LD50 (subscript 50) is the correct way of expressing this.
Author Response
The extraction method is not characterized in the current manuscript and no other references were made.
R1; Extraction method was referenced in method section.
2. The structures of the chemical constituents of Piper aduncum crude extract will be helpful as a figure
R2: Chemical structures from Piper aduncum were incorporated in manuscript.
3. It might be a good idea to elaborate more on the known effects of Piper aduncum extract
R3: More effects and traditional uses from Piper aduncum were incorporated in manuscript.
4. The paragraphs throughout the manuscript are not well developed. Authors need to make effort to develop the manuscript and use proper transition.
R4: Paragraphs in the manuscript were improved.
5. Page 6- LD50 (subscript 50) is the correct way of expressing this.
R3: this term LD50 was changed in all manuscript.
Thank you so much for your observations.
